# Research on Denoising Method for Hydroelectric Unit Vibration Signal Based on ICEEMDAN–PE–SVD

**DOI:** 10.3390/s23146368

**Published:** 2023-07-13

**Authors:** Fangqing Zhang, Jiang Guo, Fang Yuan, Yongjie Shi, Zhaoyang Li

**Affiliations:** 1Intelligent Power Equipment Technology Research Center, Wuhan University, Wuhan 430072, China; zfq@whu.edu.cn (F.Z.); 2017282070028@whu.edu.cn (Y.S.); 2017282070272@whu.edu.cn (Z.L.); 2School of Power and Mechanical Engineering, Wuhan University, Wuhan 430072, China

**Keywords:** vibration signal, improved complementary ensemble empirical mode decomposition with adaptive noise, permutation entropy, singular value decomposition, denoising

## Abstract

Vibration monitoring and analysis play a crucial role in the fault diagnosis of hydroelectric units. However, accurate extraction and identification of fault features from vibration signals are challenging because of noise interference. To address this issue, this study proposes a novel denoising method for vibration signals based on improved complementary ensemble empirical mode decomposition with adaptive noise (ICEEMDAN), permutation entropy (PE), and singular value decomposition (SVD). The proposed method is applied for the analysis of hydroelectric unit sway monitoring. Firstly, the ICEEMDAN method is employed to process the signal and obtain several intrinsic mode functions (IMFs), and then the PE values of each IMF are calculated. Subsequently, based on a predefined threshold of PE, appropriate IMFs are selected for reconstruction, achieving the first denoising effect. Then, the SVD is applied to the signal after the first denoising effect, resulting in the SVD spectrum. Finally, according to the principle of the SVD spectrum and the variation in the singular value and its energy value, the signal is reconstructed by choosing the appropriate reconstruction order to achieve the secondary noise reduction effect. In the simulation and case analysis, the method is better than the commonly used wavelet threshold, SVD, CEEMDAN–PE, and ICEEMDAN–PE, with a signal-to-noise ratio (SNR) improvement of 6.9870 dB, 4.6789 dB, 8.9871 dB, and 4.3762 dB, respectively, and where the root-mean-square error (RMSE) is reduced by 0.1426, 0.0824, 0.2093 and 0.0756, respectively, meaning that our method has a better denoising effect and provides a new way for denoising the vibration signal of hydropower units.

## 1. Introduction

Water turbine generator units play a crucial role in the global energy transformation towards green, low-carbon, and sustainable development. With the development of the economy and society, the requirements for the safety, reliability, and stability of water turbine generator units continue to increase. However, these units operate in a complex environment and face various challenges that pose significant threats to their safe and stable operation. Therefore, effective monitoring and diagnosis of the unit’s operating condition have become a research focus in the industry [1]. During the long-term operation of hydropower units, various faults and abnormalities may occur due to the impact of water flow, load changes, mechanical wear and tear, and other factors. These faults and abnormalities often lead to changes in the vibration signal of the unit. By analyzing and processing the vibration signal, key features regarding the unit status and fault information can be extracted for status evaluation and fault diagnosis of the unit. Studies have shown that nearly 80% of faults in water turbine generator units can be reflected in vibration signals [2].

Vibration signals, as an important monitoring indicator for assessing the operating status of units [3], can reflect the internal operation and fault information of the units. Through the analysis and processing of vibration signals, equipment anomalies can be detected promptly, potential failures can be predicted, downtime can be reduced, and the availability and operational efficiency of the units can be improved, ensuring their safe and stable operation [4]. Therefore, vibration signal analysis holds significant value in the fault diagnosis of water turbine generator units. However, the analysis, processing, and feature extraction of vibration signals from these units face certain challenges. Firstly, the operating status of water turbine generator units is dynamic, and vibration signals exhibit nonlinearity and nonstationarity. The frequency spectrum and amplitude vary with time, presenting complex time–frequency characteristics. This complexity makes the accurate analysis and diagnosis of vibration signals more difficult. Secondly, due to the complexity of the operating environment and the limitations of sensors in water turbine generator units, vibration signals are often subjected to various noise interferences such as mechanical noise, electromagnetic interference, and environmental noise. The presence of noise can mask or distort useful fault features, reducing the reliability of fault information in vibration signals and posing challenges for feature extraction and identification.

The accurate and reliable extraction and identification of fault characteristics in vibration signals are essential for the normal operation and maintenance of hydroelectric units. Vibration signal denoising is of great significance in the condition evaluation and fault diagnosis of hydropower units. Firstly, through the denoising process, noise interference can be effectively reduced and the quality and availability of vibration signals can be improved. Secondly, the denoising of the vibration signal of the hydropower unit can extract the characteristic information related to the operation status and fault of the unit. The vibration signal contains rich information on mechanical features, dynamic characteristics, and harmonic components, which can be used to evaluate the working status of the unit, detect abnormal operations, and predict potential faults. By denoising the vibration signal, the characteristic information can be highlighted to help engineers and technicians more accurately determine the status of the unit and take the corresponding maintenance and repair measures in time to improve the reliability and operational efficiency of the unit. Therefore, researchers have carried out a lot of research work in the field of vibration signal denoising, aiming to filter out noise interference and improve the effective extraction of real signal features.

Currently, commonly used methods for vibration signal denoising include Fourier transform, wavelet transform, empirical mode decomposition (EMD), ensemble empirical mode decomposition (EEMD), complementary ensemble empirical mode decomposition (CEEMD), complementary ensemble empirical mode decomposition with adaptive noise (CEEMDAN), variational mode decomposition (VMD), and singular value decomposition (SVD) [5,6,7,8,9]. Fourier transform is a classical linear signal analysis method suitable for analyzing stationary, regular linear signals. However, for nonstationary and nonlinear vibration signals, the Fourier transform performs poorly in extracting fault features. Wavelet transform is a method capable of analyzing nonstationary signals and allows for the analysis of signals in the time–frequency domain [6]. However, the parameter settings of the wavelet transform lack adaptability, as different signals require different wavelet basis functions, limiting its application. Empirical mode decomposition (EMD) is a relatively new time–frequency analysis algorithm that decomposes signals into several intrinsic mode functions (IMFs), enabling comprehensive analysis in the time–frequency domain. However, EMD tends to produce mode mixing and endpoint effects during signal decomposition, which limits its application in vibration signal denoising [7]. EEMD is an improved method based on EMD [8]. Although it can partially suppress mode mixing and endpoint effects caused by EMD, this method itself introduces new issues such as residual noise, decreasing the denoising effectiveness [10,11,12]. To address this, Yeh and Huang [13] proposed the CEEMD method, which effectively suppresses residual noise. However, the method has not fully addressed issues such as incomplete signal decomposition and low computational efficiency. VMD [14,15,16] effectively avoids endpoint effects seen in similar EMD methods but requires presetting the decomposition parameter K, making it difficult to achieve adaptive decomposition that meets the requirements of online automatic monitoring and analysis of vibration signals from water turbine generator units. CEEMDAN is an improved empirical mode decomposition method [17,18,19] that has certain advantages in denoising nonstationary signals. It can suppress mode mixing and endpoint effects and improve denoising effectiveness through an adaptive noise model [5,7,9]. However, it still faces issues such as residual noise and pseudo-modes [20,21,22]. Marcelo A. Colominas [23] proposed an improved denoising method called improved complementary ensemble empirical mode decomposition with adaptive noise (ICEEMDAN), which is based on CEEMDAN and addresses some of its limitations [20,21,22,24]. Unlike CEEMDAN, ICEEMDAN incorporates white noise as part of the complete noise ensemble instead of directly adding Gaussian white noise. SVD is a matrix decomposition method [25,26,27] that decomposes and transforms matrices, allowing the collected signals to be decomposed into a series of superimposed linear components. It can effectively detect subtle information variations in signals under complex backgrounds and is widely used in denoising and feature extraction [28,29,30,31,32]. In recent years, nonstationary vibration signal noise reduction via CEEMDAN–PE, ICEEMDAN–PE, and CEEMDAN–SVD has been used for the vibration feature extraction of hydropower units [7], underwater acoustic monitoring denoising [9], ball mill barrel vibration signal denoising [18], heart and brain electrical signal noise filtering [19], weld signal denoising [21], coal machine and gas signal denoising [33,34], building structure vibration signal denoising [26,35], parameter prediction [22,24,36,37] and gear-bearing fault diagnosis [31,38,39,40,41,42,43]. The initial applications of these methods have made good progress, but there is still a need to improve the vibration signal denoising of hydropower units.

To overcome the pseudo-modal problem in the CEEMDANethod and the limitations of the SVD method including matrix size limitation and information loss, as well as the unsatisfactory effect of the single denoising method of ICEEMDAN and SVD in the denoising of hydropower unit vibration signals, this study proposes a method that combines ICEEMDAN with PE and SVD for noise reduction in hydropower unit vibration signals, aiming to improve the accuracy and reliability of vibration signals and provide more effective technical support for fault diagnosis and predictions regarding hydropower units. The method processes the signal by ICEEMDAN, obtains the effective modal components (IMF), and calculates the PE value of each component. The IMF components are selected to be reconstructed according to the preset PE threshold to achieve the primary noise reduction effect [24,44,45]. Subsequently, the signal after noise reduction is processed using SVD decomposition to obtain the singular value difference spectrum, and the second noise reduction in the signal is carried out by selecting the appropriate reconstruction order; it is necessary to consider the variation in singular values and energy values, which can effectively avoid information loss to further improve the denoising effect. The results show that the proposed ICEEMDAN–PE–SVD denoising method has a higher signal-to-noise ratio and smaller root-mean-square error. Compared with the commonly used methods, this method can better retain the detailed information of the signal, effectively filter out the noise interference, and carry out more accurate and reliable fault feature extraction. Therefore, the research in this paper provides a new method for noise reduction in the vibration signal of hydropower units. In this paper, the theoretical basis, algorithm flow, simulation analysis, and example analysis of the method will be introduced in detail, which is followed by a discussion of the results.

## 2. Related Theory and Methods

### 2.1. ICEEMDAN Algorithm 

The ICEEMDAN signal processing method is an improvement over CEEMDAN [20,21,22,23,24]. Unlike the traditional EMD method, ICEEMDAN incorporates adaptive noise and a complete ensemble strategy to enhance the stability and accuracy of the decomposition process. The improved method differs from CEEMDAN in that it selects the component of white noise obtained from the EMD decomposition instead of directly adding Gaussian white noise during the decomposition process. The flowchart of the ICEEMDAN algorithm is shown in Figure 1.

Let X(t) denote the original signal to be decomposed. $E_{k}(•)$ represents the Kth IMF (intrinsic mode function) obtained through EMD decomposition. M(•) denotes the calculation of the envelope of the reconstructed signal, which yields the local mean of the reconstructed signal. $\omega^{\left(i\right)}(t)$ is the ith (i=1,2,3,⋯,I) Gaussian white noise value following a standard normal distribution N(0,1) (mean = 0, unit variance). $r_{k}(t)$ represents the residual of the kth stage. The coefficient $\beta_{k}$ represents the signal-to-noise ratio of the kth stage. When k=0, $\beta_{0}=\varepsilon_{0}\frac{\sigma (x(t))}{\sigma (E_{1}(\omega^{\left(i\right)}(t)))}$, and when k≥1, $\beta_{k}=\varepsilon_{0}\sigma (r_{k}(t))$ (where $\varepsilon_{0}$ is a predetermined amplitude parameter and σ(•) is the standard deviation operator). The notation $\langle •\rangle$ denotes the operator for calculating the average value [21,22].

The steps of the ICEEMDAN decomposition algorithm are as follows [23]:

Step 1. Add white noise to the original signal X(t) according to Equation (1):(1)$$X^{\left(i\right)}(t)=X(t)+\beta_{0}E_{1}(\omega^{\left(i\right)}(t))$$

Step 2. Decompose the obtained signal $X^{\left(i\right)}(t)$ using the EMD method, calculate the local mean $M(X^{\left(i\right)}(t))$ of each component as described in Equation (2), and obtain $r_{1}(t)$ (the residual of the k=1 stage):(2)$$r_{1}(t)=\langle M(X^{\left(i\right)}(t))\rangle$$

Step 3. Calculate the k=1 component of the original signal $IMF_{1}(t)$ as described in Equation (3):(3)$$IMF_{1}(t)=X(t)-r_{1}(t)$$

Step 4. Add white noise to the residual of the first stage $r_{1}(t)$, calculate the local mean of the signal, and obtain $r_{2}(t)$ (the residual of the k=2 stage) using Equation (4):(4)$$r_{2}(t)=\langle M(r_{1}(t)+\beta_{1}E_{2}(\omega^{\left(i\right)}(t)))\rangle$$

Step 5. Calculate the k=2 component of the original signal $IMF_{2}(t)$ as described in Equation (5):(5)$$IMF_{2}(t)=r_{1}(t)-r_{2}(t)$$

Step 6. When k=3,4,5,⋯K, calculate $r_{k}(t)$(the residual of the kth stage) using Equation (6):(6)$$r_{k}(t)=\langle M(r_{k-1}(t)+\beta_{k-1}E_{k}(\omega^{\left(i\right)}(t)))\rangle$$

Step 7. Calculate the kth component of the original signal as described in Equation (7):(7)$$IMF_{k}(t)=r_{k-1}(t)-r_{k}(t)$$

Step 8. Return to Step 6 and calculate the next k.

Step 9. The original signal, after being decomposed by ICEEMDAN, can be represented as:(8)$$X(t)=\sum_{k=1}^{K}IMF_{k}(t)+r_{k}(t)$$

### 2.2. PE Algorithm

PE is a nonlinear analysis method used to detect randomness and dynamic transitions in time series data [19,24,44,45]. It is an indicator commonly used to measure the complexity and nonlinear characteristics of signals and is widely employed in the analysis of time series and complex systems. The calculation of PE is based on the arrangement of the signal, where the signal sequence is sorted to obtain different permutations. The occurrence frequency of each permutation is then counted, and the probability distribution of each permutation is calculated. PE reflects the irregularity of the signal over time, where higher values indicate greater complexity and nonlinear features in the signal. It is highly sensitive to signal transitions and can quantitatively assess the presence of random noise in a signal sequence. The calculation process of PE is illustrated in Figure 2.

Let X(t) be a time series of length N: {X(t),t=1,2,3.⋯,N},m is the embedding dimension, τ is the time delay, and K=N−(m−1)τ is the number of reconstructed components. The calculation steps for PE are as follows [19,24,44,45]:

Step 1. Perform phase-space reconstruction on the time series X(t) to obtain the m-dimensional delay vector sequence, denoted as matrix Z:(9)$$Z=\left(\begin{array}{cccc} x(1) & x(1+\tau ) & \cdots & x(1+(m-1)\tau )\\ \vdots & \vdots & & \vdots \\ x(j) & x(j+\tau ) & \cdots & x(j+(m-1)\tau \\ \vdots & \vdots & & \vdots \\ x(K) & x(K+\tau ) & \cdots & x(K+(m-1)\tau \end{array}\right)$$

Step 2. Each row of the matrix represents a reconstructed component. Rearrange each component (x(j),x(j+τ),⋯,x(j+(m−1)τ) in matrix Z in ascending order according to their values:(10)$$x(t+(j_{1}-1)\tau )\leq x(t+(j_{2}-1)\tau )\leq \cdots \leq x(t+(j_{m}-1)\tau )$$
where t represents the column index, and $j_{1},j_{2},\cdots ,j_{m}$ represent the positions of each element in the reconstructed component.

Thus, for any time series X(t), the matrix Z obtained through reconstruction will have a set of symbolic sequences.
(11)$$S(l)=(j_{1},j_{2},\cdots ,j_{m})$$
where l=1,2,⋯,k, and 1≤k≤m!, the m-dimensional phase space maps different sequences of symbols with a total of m! possibilities of permutations, and S(l) is one of them.

Step 3. Calculate the probability of occurrence for each symbolic sequence, denoted as $P_{1},P_{2},\cdots ,P_{k}$. The probability is calculated as the count of occurrences of each symbolic sequence divided by m! (the total count of different symbolic sequences). Define the PE as the Shannon entropy of the probability distribution, which measures the uncertainty or randomness of the time series:(12)$$H_{P}(m)=-\sum_{j=1}^{k}P_{j}\ln(P_{j})$$

To facilitate comparison, the PE is often normalized by dividing it by the maximum value it can achieve, denoted as $H_{P}{\left(m\right)}_{\max}=\ln(m!)$. This normalization yields a normalized PE $H_{P}$:(13)$$H_{P}=\frac{H_{P}(m)}{\ln(m!)}$$
where $H_{P}\in \left[0,1\right]$. The value of $H_{P}$ reflects the randomness or irregularity of the time series, and it is positively correlated with the degree of randomness. The variation in $H_{P}$ amplifies the subtle details of the time series.

### 2.3. SVD Algorithm

SVD is a matrix decomposition method that decomposes a matrix into the product of three matrices: an orthogonal matrix, a diagonal matrix, and the transpose of another orthogonal matrix. SVD is commonly used in signal processing for tasks such as data dimensionality reduction, signal denoising, and feature extraction [25,26,27,28,29,30,31,32]. In signal denoising, SVD aims to eliminate noise by constructing a matrix that contains the signal information and then decomposing this matrix into a series of singular values and corresponding singular vectors representing the time–frequency subspaces. This approach helps retain useful information related to the faulty signal and finds extensive applications in signal analysis and processing [43].

Let X(t) be a time series of length N: {X(t),t=1,2,3.⋯,N}. The specific steps of SVD are as follows:

Step 1. Phase-space reconstruction is carried out by constructing an m×n order Hankel matrix (H) for the signal to be decomposed.
(14)$$H=\left[\begin{array}{cccc} x(1) & x(2) & \cdots & x(n)\\ x(2) & x(3) & \cdots & x(n+1)\\ \vdots & \vdots & \ddots & \vdots \\ x(m) & x(m+1) & \cdots & x(N) \end{array}\right]$$
where in Equation (14), N=m+n−1, N>m≥n>1, and $H\in R^{m\times n}$. The number of rows, m, and the number of columns, n, can be determined based on the following principles: when the length N is even, we set $m=\frac{N}{2}+1$ and $n=\frac{N}{2}$; when the length *N* is odd, we have $n=m=\frac{N+1}{2}$.

Step 2. Perform SVD decomposition on the matrix H.
(15)$$H=USV^{T}=U\left[\begin{array}{cc} \Lambda & 0\\ 0 & 0 \end{array}\right]V^{T}$$

In Equation (15), $U\in R^{m\times m}$ and $V\in R^{n\times n}$ are orthogonal matrices, representing the left and right singular vectors, respectively. $S\in R^{m\times n}$ is a diagonal matrix with its diagonal elements being the singular values of H. Let Λ denote the diagonal matrix formed by arranging the singular values $\lambda_{i}$ in descending order, given by $\Lambda =\mathrm{diag}(\lambda_{1},\lambda_{2},\cdots ,\lambda_{q})$, where q=min(m,n) and $\lambda_{1}\geq \lambda_{2}\geq \cdots \geq \lambda_{q}$.

Step 3. Compute the differences between adjacent singular values, resulting in a sequence of differential singular values $b_{i}$. Choosing the appropriate order of singular values is crucial for signal reconstruction, and the differential singular value spectrum effectively captures the singular value variations in noisy signals.
(16)$$b_{i}=\lambda_{i}-\lambda_{i+1}$$

In Equation (16), i=1,2,3,⋯,q and $\lambda_{q+1}=0$.

k exists such that $b_{k}$ is the peak point and there is no abrupt change in the singular value after the values of k and $b_{k}$ gradually approach 0. In the case of the singular value sequence, the maximum peak represents the boundary between the noise and useful signal. When k=1, it indicates the presence of a DC component in the signal; we assign the second largest peak to k. On the other hand, when k>1, it suggests that the signal does not contain a DC component or that the DC component is smaller than the AC component [46]. In a noisy signal segment, the useful signal only contributes to the first singular values of the reconstructed matrix, while the noise uniformly contributes to the singular values [29].

Step 4. After determining the value of k, the singular values beyond are set to zero, resulting in a diagonal matrix where $\Lambda_{0}=\mathrm{diag}(\lambda_{1},\lambda_{2},\cdots ,\lambda_{k},\cdots ,\lambda_{q})$, $\lambda_{k+1}=\lambda_{k+2}=\cdots =\lambda_{q}=0$, and $S_{0}=\left[\begin{array}{cc} \Lambda_{0} & 0\\ 0 & 0 \end{array}\right]$, which is used to reconstruct the signal through SVD [28]. By analyzing the noise test results, we can obtain the denoised signal represented by the matrix $H_{0}$.
(17)$$H_{0}=US_{0}V^{T}=U\left[\begin{array}{cc} \Lambda_{0} & 0\\ 0 & 0 \end{array}\right]V^{T}$$

## 3. ICEEMDAN–PE–SVD-Based Vibration Signal Denoising for Hydropower Units

The proposed denoising method, ICEEMDAN–PE–SVD, combines the advantages of the aforementioned three algorithms to achieve dual denoising of turbine vibration signals under strong noise and complex electromagnetic interference backgrounds. The flowchart of the denoising method based on ICEEMDAN–PE–SVD for hydroelectric turbine vibration signals is shown in Figure 3. The detailed steps are shown in Algorithm 1.
**Algorithm 1.** Denoising method based on ICEEMDAN–PE–SVD for hydroelectric turbine vibration signals.**Input**: Original signal x(t).**Output**: Denoised signal $x^{\prime }(t)$.1: According to Equations (1)–(8), the ICEEMDAN decomposition of the original signal x(t) is calculated to obtain a set of intrinsic mode functions ($IMF_{k}(t)$) and a residue component ($r_{k}(t)$). $X(t)\to \sum_{k=1}^{K}IMF_{k}(t)+r_{k}(t)$2: The PE of the resulting IMF components is calculated according to Equations (9)–(13). In [34], Brandt et al., after a lot of experiments and projections, recommended that the statistical results have high reasonableness when the embedding dimension m is taken from 3 to 7, and the delay time has less influence on the calculation of the PE. Therefore, in this paper, we chose the number of embedding bits of m=5, and the delay time of τ=1. $IMF_{k}\to H_{P}(IMF_{k})$3: The normalized PE threshold is set to 0.3 according to the results of multiple simulation experiments and combined with the PE calculation principle. **for** $IMF_{k}$ in $IMF_{K}$
**do****if** $H_{P}(IMF_{k})\leq 0.3$, $IMF_{k}$ is selected as the valid IMF component **return** all the valid IMF components.4: The signal $Sig_{iceemdan-pe}$ is obtained by reconstructing it according to all the valid IMF components. $IMF\to Sig_{iceemdan-pe}$5: The SVD decomposition of $Sig_{iceemdan-pe}$ occurs according to Equations (14) and (15).6: Calculate the difference spectrum of singular values obtained from the decomposition according to Equation (16).7: Combined with the variation trend of the difference spectrum, an appropriate singular value order k is selected to reconstruct the characteristic matrix $H_{0}$, and then the matrix $H_{0}$ is converted to the denoised signal $x^{\prime }(t)$.8: **return** $x^{\prime }(t)$. This achieves a double denoising effect and forms the final denoised signal, finishing the denoising process of the vibration signal of the hydropower unit.

## 4. Simulation Analysis

### 4.1. Construction of the Simulation Signal

To verify the effectiveness of the vibration signal denoising method based on ICEEMDAN–PE–SVD, the oscillation signal of hydropower units was selected for simulation analysis. The pendulum of the hydropower unit is mainly affected by mechanical excitation and hydraulic excitation, and the mechanical excitation is generally dominated by a medium frequency (1, 2, or 3 times the rotational frequency), while the hydraulic excitation is generally dominated by a low frequency (0.2~0.45 times the rotational frequency). Choosing the rotational frequency as 1.25Hz, we used a frequency that is 1, 2, 3, 4, 0.2, 0.3, or 0.45 times the rotational frequency of the signal superposition and simulated the mechanical vibration and hydraulic vibration at the same time, so that the simulated signal is constructed as follows [47]:(18)$$f(t)=\sum_{i=1}^{7}A_{i}\sin2\pi f_{i}t$$

In Equation (18), $A_{1}\text{\textasciitilde{}}A_{7}$ represent amplitudes of 20 μm, 4.5 μm, 2.55 μm, 1.5 μm, 0.4 μm, 0.3 μm, and 0.2 μm, respectively. $f_{1}\text{\textasciitilde{}}f_{7}$ represent frequencies of 1.25 Hz, 2×1.25Hz, 3×1.25Hz, 4×1.25Hz, 0.2×1.25Hz, 0.3×1.25Hz, and 0.45×1.25Hz. The sampling frequency was set to 1000Hz. The simulated original signal without noise is shown in Figure 4. The simulated noisy signal was obtained by adding Gaussian white noise with a signal-to-noise ratio of 20dB, as shown in Figure 5.

### 4.2. Noise Reduction in Simulated Signals by ICEEMDAN–PE–SVD

When performing ICEEMDAN decomposition on the simulated signal, it is necessary to use reasonable settings including the noise amplitude coefficient $\varepsilon_{0}$ that is added to the original signal, the average number of times I for measuring the signal, the total number of times M for performing ICEEMDAN and the maximum allowed number of iterations MaxIter.

Combined with the findings from [23], after several experiments; setting the parameters as $\varepsilon_{0}=0.2$, I=100, M=1000, and MaxIter=1000; and then performing ICEEMDAN decomposition on the synthesized simulated signal, 10 IMF components and 1 trend component R were obtained, as shown in Figure 6. The PE of these components was calculated and the results are shown in Table 1.

According to the principle of PE combined with the results of several simulation experiments, the final threshold value of PE was set to 0.3, and the IMF components (IMF6~IMF10) with a threshold value less than 0.3 were selected and reconstructed with the residual R. To make it more convenient to see the effect of reconstructing the signal by different number of components after ICEEMDAN decomposition during simulation analysis, the signals that were reconstructed with different numbers of components after ICEEMDAN decomposition are shown in Figure 7.

From Figure 7, it can be seen that the reconstructed signals containing IMF1, IMF2, IMF3, IMF4, and IMF5 components have obvious noise interference, while the reconstructed signals with IMF6~IMF10 and R components are smoother and closer to the original simulated signals, which are consistent with the structure of the components selected by the PE threshold. Therefore, the IMF6~IMF10 components are classified as valid components, and the reconstructed waveforms and spectrograms are shown in Figure 8.

From Figure 8, we can see that the waveform and spectrum after one reconstruction of ICEEMDAN–PE show a certain trend compared with the original noise-free waveform and spectrum, but the figure does not show the simulated pendulum signal clearly, while the spectrum can also show the characteristic frequency but there is still residual noise.

Then, the reconstructed signal was processed with SVD decomposition, and the decomposed singular value difference spectrum was obtained, as shown in Figure 9 below.

As shown in Figure 9, when the singular value order k=12, the singular value difference spectrum and the singular value energy are relatively large; thus, a singular value of k>12 does not produce abrupt changes. According to the principle of the singular value difference spectrum, and considering the changes in the singular value and its energy value, the final order of reconstruction chosen in this paper is k=12, i.e., the components with a singular value less than 12 are selected for reconstruction, as shown in Figure 10, which is the waveform and spectrum after ICEEMDAN–PE–SVD noise reduction.

As shown in Figure 10, after the multiple noise reductions by ICEEMDAN–PE–SVD, the waveforms and spectrograms can be compared with the original noise-free waveforms and spectrograms. The waveforms and spectrograms are basically the same, and the main characteristic frequencies are completely extracted. The bottom noise is completely removed, indicating that the method can effectively remove the noise and maintain the integrity of the original signal.

### 4.3. Comparative Analysis of Related Denoising Method Indices

To facilitate a quantitative comparison of the denoising performance of different methods, the signal-to-noise ratio (SNR), root-mean-square error (RMSE), and mean absolute error (MAE) are defined below [32,36], and the larger the SNR and the smaller the RMSE and MAE, the better the denoising effect.

(1)Signal-to-noise ratio (SNR):


(19)
$$\mathrm{SNR}=10\mathrm{lg}\frac{\sum_{i=1}^{N}X_{i}^{2}}{\sum_{i=1}^{N}{\left[Y_{i}-X_{i}\right]}^{2}}$$


(2)Root-mean-square error (RMSE):


(20)
$$\mathrm{RMSE}=\sqrt{\frac{1}{N}\sum_{i=1}^{N}[Y_{i}-X_{i}{]}^{2}}$$


(3)Mean absolute error (MAE):


(21)
$$\mathrm{MAE}=\frac{1}{N}\sum_{i=1}^{N}\left|Y_{i}-X_{i}\right|$$


In Equation (19) to Equation (21): N is the number of sampling points; $X_{i}$ is the original un-noised simulated signal; and $Y_{i}$ is the denoised signal.

To verify the effectiveness of the ICEEMDAN–PE–SVD denoising method proposed in this paper in the denoising of the oscillation signal of hydropower units, the wavelet threshold denoising method, the CEEMDAN–PE [19] denoising method, the ICEEMDAN denoising method, and the ICEEMDAN–PE–SVD denoising method proposed in this paper were used to denoise the simulated additive noise signal. The denoising effect of these different methods was compared and analyzed by using the denoising waveforms and denoising performance indices.

The selection of parameters in the wavelet threshold denoising process has a great influence on the denoising effect of the original signal. In this paper, according to the method of determining wavelet threshold denoising parameters in [18], the wavelet threshold denoising parameters of the simulated noise-added signal are set as follows: ‘sym10′ as the wavelet basis function, a decomposition layer of 4, ’heursure’ as the threshold guidelines, and a hard threshold function. The waveforms after denoising are shown in Figure 11 and Figure 12, and the performance indices of the denoising effect are shown in Table 2.

By comparing Figure 11 and Figure 12 with the simulated un-noised signal in Figure 4, it is found that wavelet thresholding and CEEMDAN denoising can filter out most of the noise in the simulated denoised signal, but the burr phenomenon in the denoised signal waveform is more obvious and the denoising effect is less satisfactory.

Comparing Figure 8 with the waveform in Figure 4, it can be found that the ICEEMDAN–PE denoising signal is achieved by denoising the simulated additive noise signal. As can be seen from Figure 10, ICEEMDAN–PE–SVD denoising is a process of noise reduction in the mixed signal-noise components discarded by the ICEEMDAN–PE filtering and denoising method, so that some effective features of the original signal can be retained while keeping the waveform smooth.

According to the denoising performance evaluation criteria and by comparing the data in Table 2, it can be seen that the ICEEMDAN–PE–SVD denoising method has the best denoising effect, and compared with the wavelet threshold, SVD, CEEMDAN–PE, and the ICEEMDAN–PE denoising method, the signal-to-noise ratio (SNR) is improved by 6.9870 dB, 4.6789 dB, 8.9871 dB, and 4.3762 dB, respectively, and the root-mean-square error (RMSE) is reduced by 0.1425, 0.0824, 0.2092 and 0.0755, respectively.

In summary, by comparing and analyzing the waveform characteristics and denoising performance evaluation indices of denoised signals of different denoising methods, it was found that the combined ICEEMDAN–PE–SVD denoising method is significantly better than the other four methods.

## 5. Case Analysis

To verify the effectiveness of the proposed ICEEMDAN–PE–SVD denoising method in the process of denoising the vibration signal of a hydropower unit, an example of the pendulum monitoring of unit 1 of a hydropower station was further selected for analysis. The turbine model is HLA773a-LJ-200, with a rated head of 68 m. The Online monitoring device for unit vibration and swing is shown in Figure 13, which has an integrated NI high-speed acquisition card and an eddy current displacement sensor, CWY-DO-20Q08-50V.

The speed of the unit in the process of hydro-generator pendulum signal acquisition was 250r/min, the sampling frequency $f_{s}$ of the test was 2048Hz, the water head was 65.8m and 6000 points of the sampled data were intercepted for analysis to ensure that the characteristic parameters in the signal processing analysis can fully and truly reflect the actual working conditions.

The measured signal waveform of the *X*-direction swing of the upper guide of the hydroelectric unit is shown in Figure 14, which contains a large number of burrs. The ICEEMDAN decomposition was first performed on this measured signal to obtain 10 IMF components and 1 trend component R. The PE value of each component was calculated, and the results are shown in Table 3. Similarly, the six components (IMF5-IM10) with a PE value less than 0.3 were selected as the effective information components for reconstruction, and then the SVD algorithm (singular value order k=7) performs the secondary noise reduction to complete the denoising of the measured signal of the upper conductivity of the hydropower unit. It can be seen from Figure 15 that the noise in the measured signal of the upper guidance swing degree has been effectively filtered.

Since the original, pure pendulum signal value cannot be obtained from the measured signal and the SNR and RMSE cannot be calculated directly, the noise rejection ratio (NRR) before and after signal denoising is defined below to characterize the prominence of the effective signal after denoising. The larger the value, the more prominent the effective signal after denoising [15].
(22)$$NRR=10\mathrm{lg}10(\sigma_{1}^{2}-\sigma_{2}^{2})$$

In Equation (22), $\sigma_{1}^{2}$ and $\sigma_{2}^{2}$ denote the variance in the signal before and after denoising, respectively. The noise suppression examples of the four denoising methods after processing the measured signal of the pendulum degree are shown in Table 4. From Table 4, we can see that ICEEMDAN–PE–SVD denoising has the highest NRR value and the best filtering effect, whereas the denoising with ICEEMDAN–PE and CEEMDAN–PE is in second, and wavelet threshold denoising is the worst.

The measured signal and denoised signal waveforms of the *Y*-direction swing of the lower guide and the *X*-direction swing of the water guide of the hydropower unit are shown in Figure 16, Figure 17, Figure 18 and Figure 19, respectively. ICEEMDAN decomposition was performed on these two signals to obtain the IMF component and trend component. The entropy value of each component was calculated, and the results are shown in Table 3. According to the PE threshold, the division of components was selected to reconstruct, and then the secondary denoising was performed on the reconstructed signal using the SVD algorithm, which finally completes the denoising of the measured signals of the *Y*-direction of the lower guide and the *X*-direction of the water guide of the hydropower unit, as shown in Figure 17 and Figure 19, respectively.

**Table 3 sensors-23-06368-t003:** PE of each component for case analysis by ICEEMDAN.

BSS	F1	F2	F3	F4	F5	F6	F7	F8	F9	F10	R
UG	0.972	0.748	0.535	0.402	0.287	0.225	0.163	0.148	0.148	0.139	0.017
LG	0.964	0.784	0.55	0.329	0.269	0.189	0.163	0.15	0.15	–	0.135
WG	0.964	0.75	0.536	0.368	0.282	0.208	0.177	0.163	0.151	0.149	0.146

In Table 3 and Table 4, BSS denotes the bearing swing signal; UG denotes the upper guide bearing; LG denotes the lower guide bearing; and WG denotes the water guide bearing.

**Table 4 sensors-23-06368-t004:** The noise rejection ratio of de-noised swing signals by different methods.

Denoising Method	UG NRR/dB	LG NRR/dB	WG NRR/dB
Wavelet Threshold	2.1569	3.2387	4.7788
SVD	11.1864	20.2995	16.5618
CEEMDAN–PE	10.1356	12.1863	15.7854
ICEEMDAN–PE	6.6525	16.8692	14.1723
**ICEEMDAN–PE–SVD**	**11.7286**	**20.311**	**16.6323**

From the above comparison of the figures, it can be seen that the method based on ICEEMDAN–PE–SVD can effectively remove a large amount of background noise contained in the measured signals of the lower guide and water guide pendulums of hydroelectric units. The noise rejection ratios of five different denoising methods for the down–conductor and hydro-conductor pendulum signals were also calculated. From Table 4, it can be seen that the ICEEMDAN–PE–SVD denoising method has the best denoising effect.

## 6. Conclusions

In this paper, an ICEEMDAN–PE–SVD-based denoising method for the vibration signals of hydroelectric units was proposed. The method can be used to denoise the simulated and three measured signals of hydropower unit swing, and the following conclusions are obtained after comparing the waveform differences and the magnitude of denoising performance indices with the methods of wavelet threshold denoising, CEEMDAN–PE denoising, and ICCEMDAN–PE denoising in comparison tests:(1)Through simulation tests, the ICEEMDAN–PE–SVD method proposed in this paper, after the double-noise reduction process, obtains a root-mean-square error as low as 0.1152 and the signal-to-noise ratio is improved to 42.0941, which maximizes the noise elimination while retaining the useful information within the fault signal. The method has a good noise reduction and pulse effect, and avoids modal mixing in the EMD decomposition process and the pseudo-modal problem of CEEMDAN decomposition.(2)Through the case analysis of the oscillation data of the measured hydro-generator unit’s upper guide in the X-direction, the lower guide in the Y-direction, and the water guide in the X-direction, it was found that the method can effectively reduce the noise of the measured unit data and extract the characteristic frequency of the vibration signal more accurately so that the cause of the unit vibration can be judged by the frequency. The denoising effect of the measured signal was better than that of the traditional method, as it can effectively filter out the noise components and provide a powerful tool for the online monitoring of equipment vibration signals.(3)The research results of this paper can also be widely applied to signal denoising and feature extraction of high-safety equipment in nuclear power, power grids, the petrochemical industry, and other industries.

## Figures and Tables

**Figure 1 sensors-23-06368-f001:**
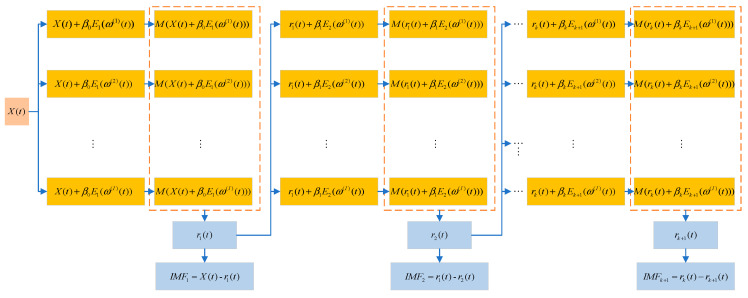
ICEEMDAN algorithm flow chart.

**Figure 2 sensors-23-06368-f002:**
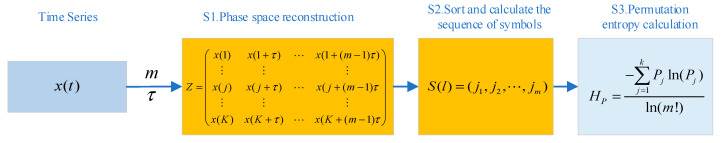
PE algorithm flow chart.

**Figure 3 sensors-23-06368-f003:**
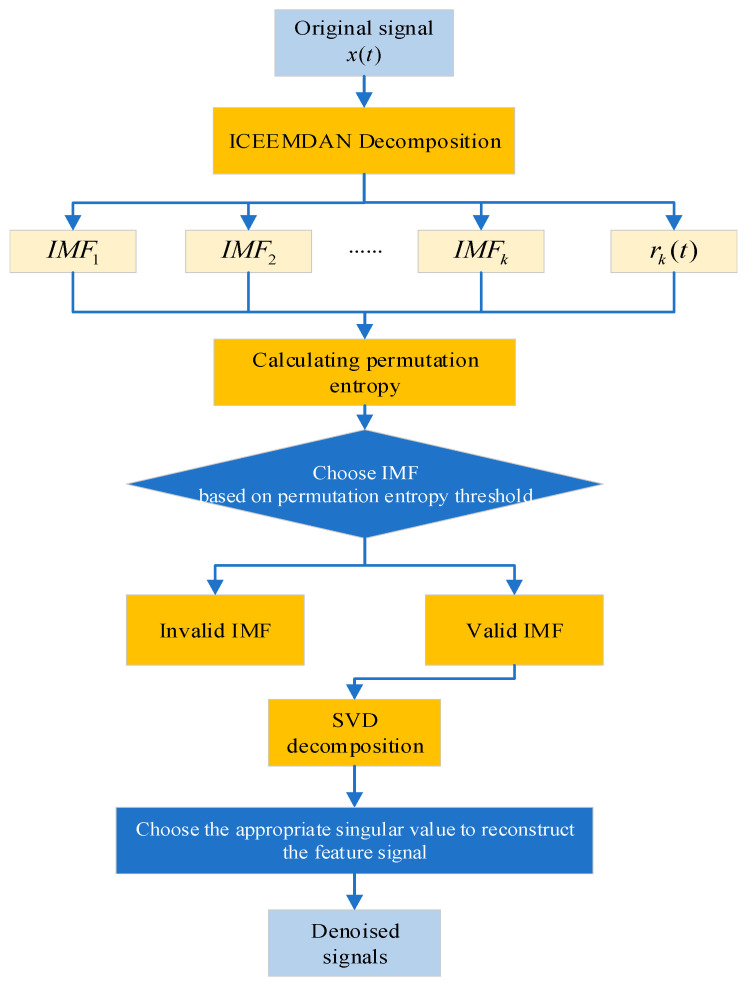
Process of signal denoising method based on ICEEMDAN–PE–SVD.

**Figure 4 sensors-23-06368-f004:**
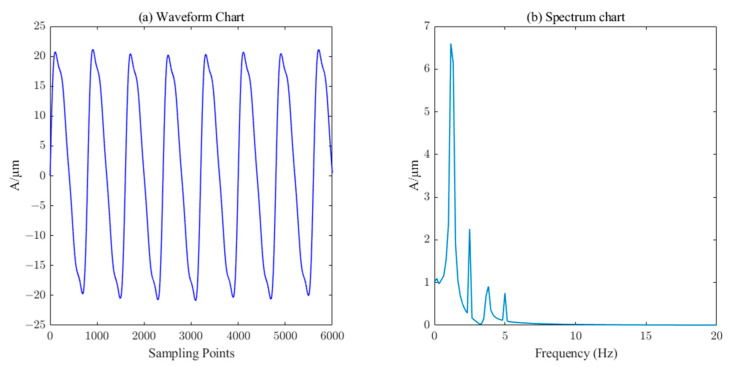
Waveform and spectrum chat of simulation signal without noise.

**Figure 5 sensors-23-06368-f005:**
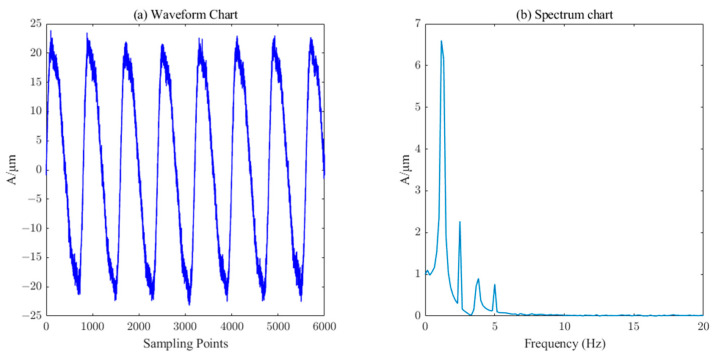
Waveform and spectrum chat of simulation signal with noise.

**Figure 6 sensors-23-06368-f006:**
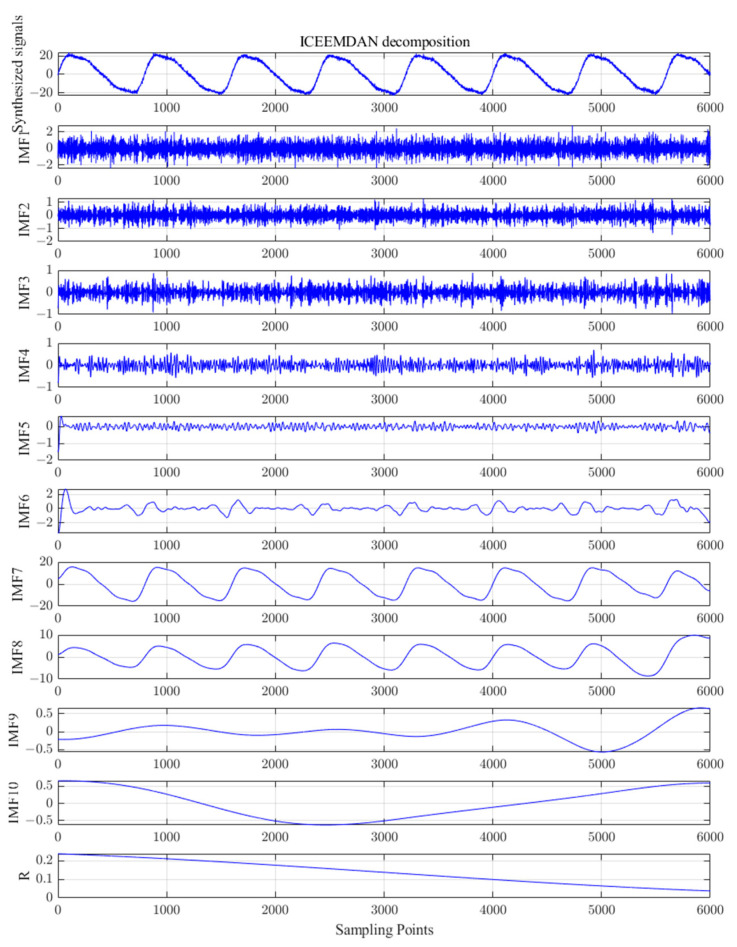
ICEEMDAN decomposition of the simulation signals.

**Figure 7 sensors-23-06368-f007:**
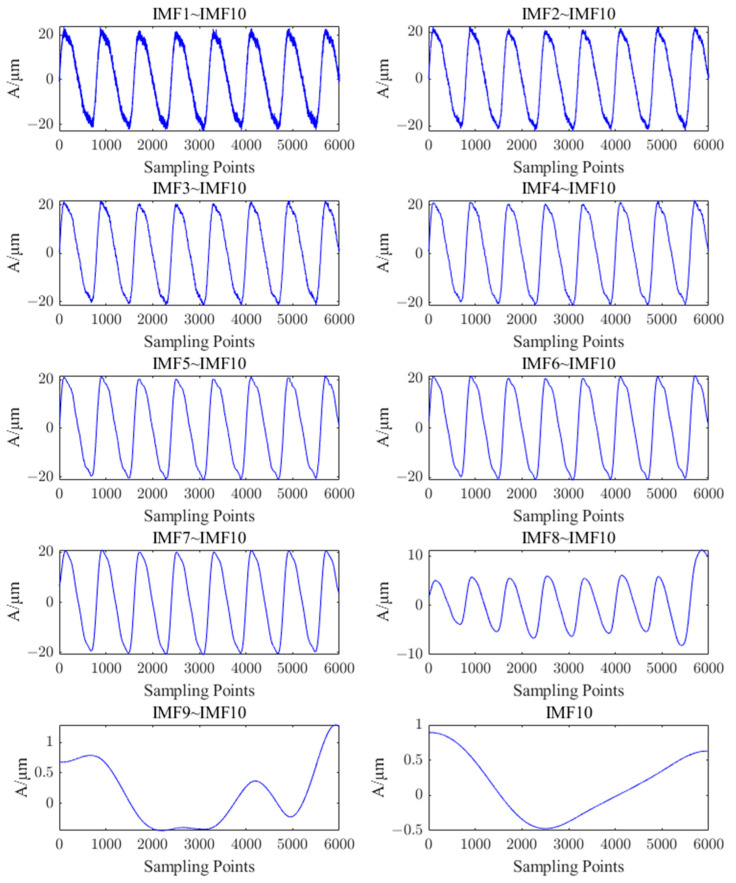
Reconstructed signals with different numbers of IMF components.

**Figure 8 sensors-23-06368-f008:**
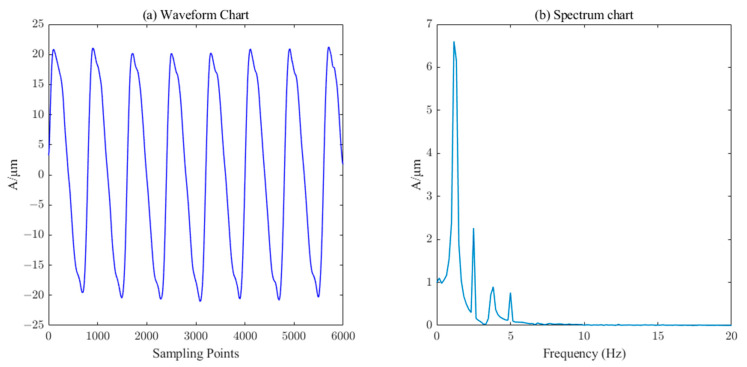
Waveform and spectrum chat by ICEEMDAN–PE.

**Figure 9 sensors-23-06368-f009:**
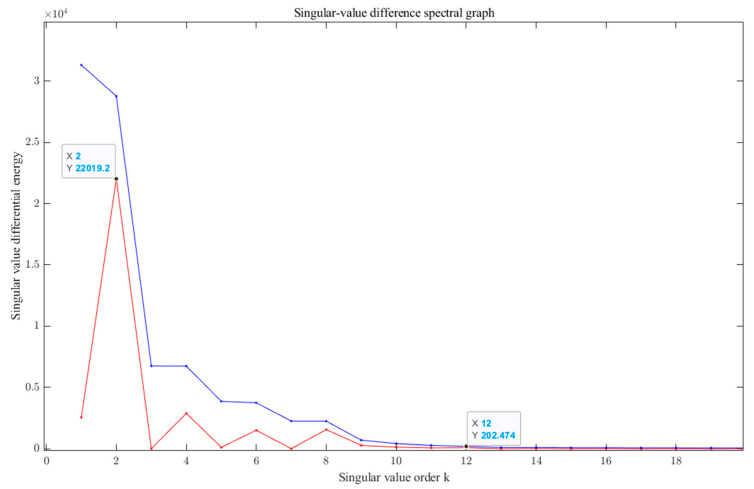
Singular value difference spectral graph of SVD. Red line means singular value; blue line means singular value difference.

**Figure 10 sensors-23-06368-f010:**
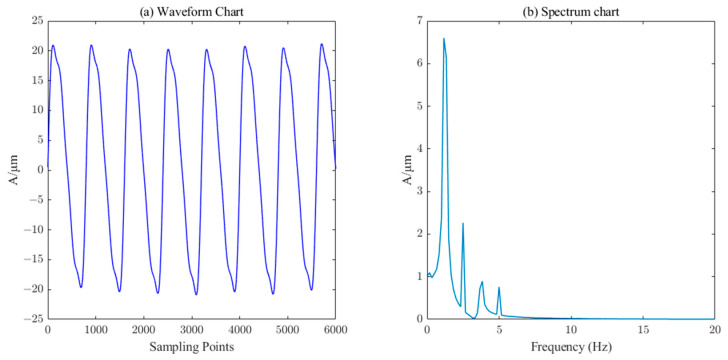
Waveform and spectrum chart of ICEEMDAN–PE–SVD noise reduction.

**Figure 11 sensors-23-06368-f011:**
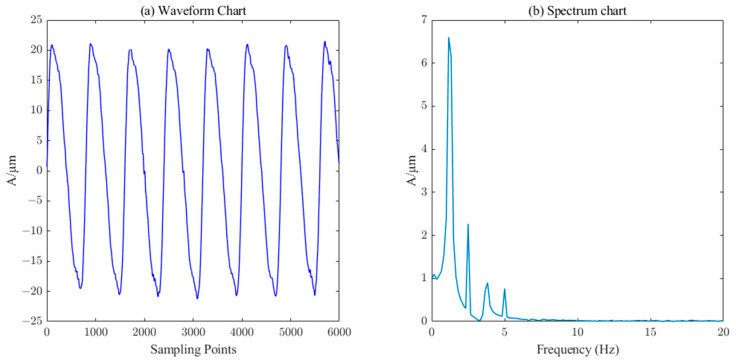
Waveform and spectrum chart after wavelet threshold noise reduction.

**Figure 12 sensors-23-06368-f012:**
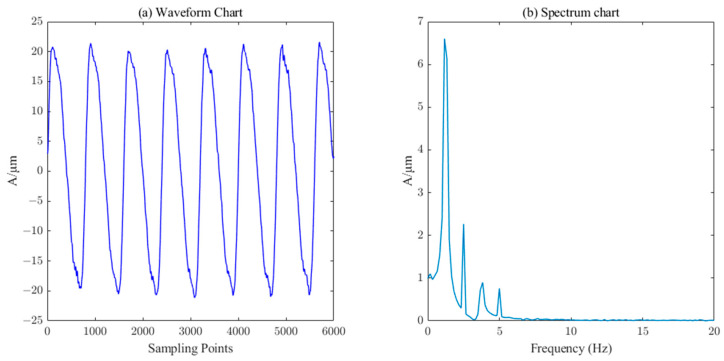
Waveform and spectrum chart by CEEMDAN–PE.

**Figure 13 sensors-23-06368-f013:**
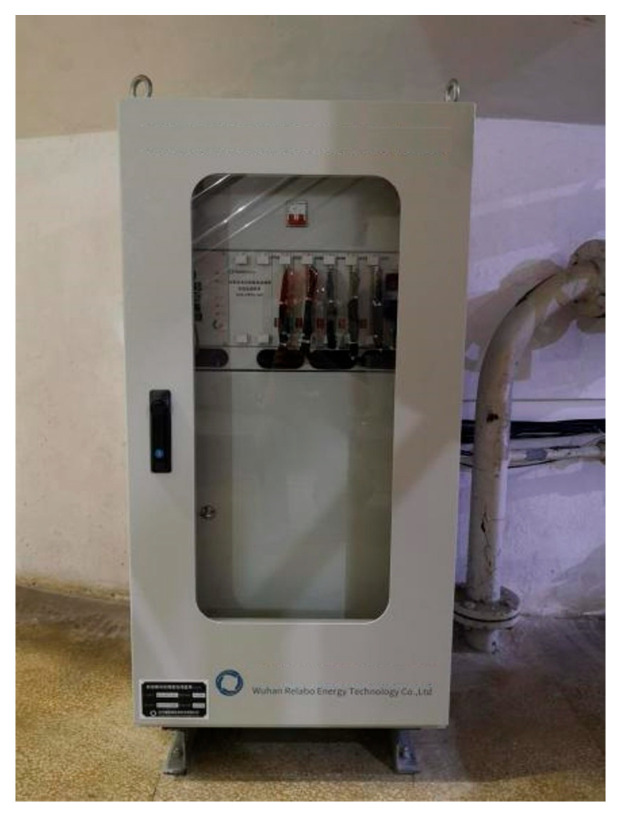
Online monitoring device for unit vibration and swing.

**Figure 14 sensors-23-06368-f014:**
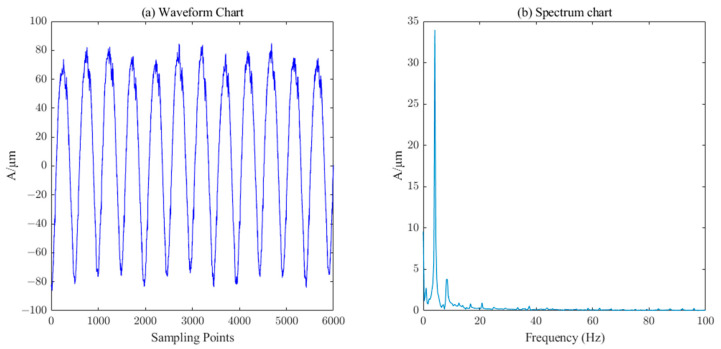
Waveform and spectrum chart of real signals of upper–guide–bearing swing.

**Figure 15 sensors-23-06368-f015:**
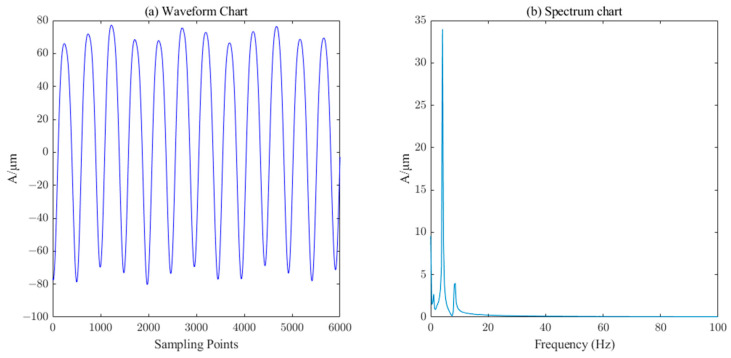
Waveform and spectrum chart of de-noised upper–guide–bearing swing signals by ICEEMDAN–PE–SVD (singular value order k=7.

**Figure 16 sensors-23-06368-f016:**
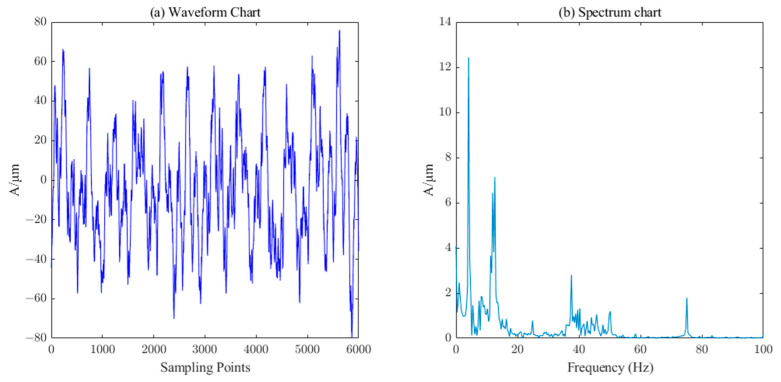
Waveform and spectrum chart of real signals of lower–guide–bearing swing.

**Figure 17 sensors-23-06368-f017:**
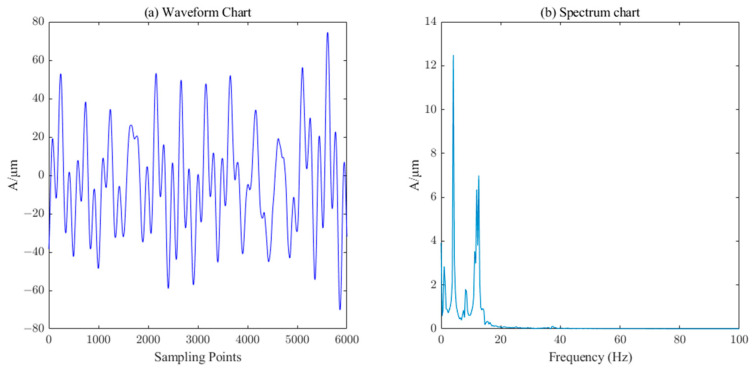
Waveform and spectrum chart of de-noised lower–guide–bearing swing signals by ICEEMDAN–PE–SVD (singular value order k=15).

**Figure 18 sensors-23-06368-f018:**
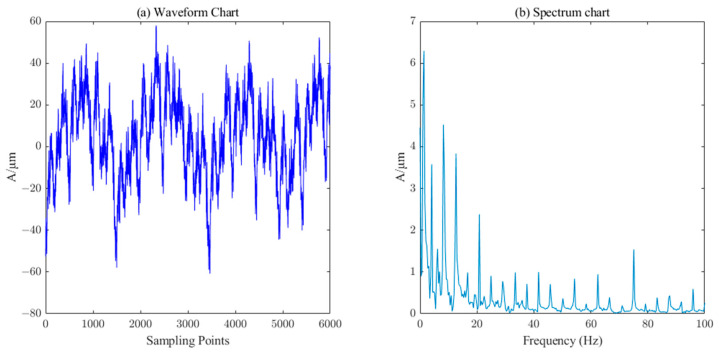
Waveform and spectrum chart of real signals of water–guide–bearing swing.

**Figure 19 sensors-23-06368-f019:**
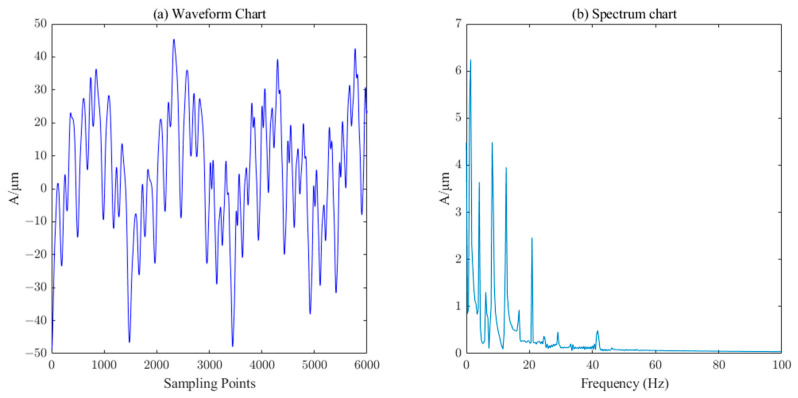
Waveform and spectrum chart of de-noised water–guide–bearing swing signals by ICEEMDAN–PE–SVD (singular value order k=20).

**Table 1 sensors-23-06368-t001:** PE of each component for simulation analysis by ICEEMDAN.

IMF_1_	IMF_2_	IMF_3_	IMF_4_	IMF_5_	IMF_6_	IMF_7_	IMF_8_	IMF_9_	IMF_10_	R
0.9375	0.7752	0.5679	0.4093	0.309	0.212	0.1362	0.1489	0.1498	0.1436	0.0019

**Table 2 sensors-23-06368-t002:** Denoising performance of different methods.

Denoising Method	SNR/dB	RMSE	MAE
Wavelet Threshold	35.1071	0.2578	0.2027
SVD	37.4152	0.1976	0.1424
CEEMDAN–PE	33.1070	0.3245	0.2504
ICEEMDAN–PE	37.7179	0.1908	0.1390
**ICEEMDAN–PE–SVD**	**42.0941**	**0.1152**	**0.0909**

## Data Availability

The raw/processed data cannot be shared at this time. Due to the nature of this research, participants of this study did not agree for their data to be shared publicly.

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
