# Peer review of "Research on Denoising Method for Hydroelectric Unit Vibration Signal Based on ICEEMDAN–PE–SVD"

_sensors, 2023, doi:10.3390/s23146368_

Round 1
Reviewer 1 Report
This manuscript presented an ICEEMDAN-PE-SVD-based denoising method for the vibration signals of hydroelectric units, denoised the simulated and three measured signals of hydropower unit swing respectively. Although presented work is interesting and provides novel information for the Journal readers, some remarks are given to improve the quality of the paper:
1. In the introduction, more industrial application background of this kind of the vibration signals of hydroelectric units could help readers to clearly understand the purpose of this design.
2. Why do you choose the ICEEMDAN-PE-SVD to obtain the noise components? I suggest you give some literature to show its advantages.
3. Are there any similar researches that use the ICEEMDAN-PE-SVD to address the above problem? If there are some, please emphasize the merits and the difference of your method. I suggest the authors should give more explanations about the new contribution of this work.
4. The objects and platforms for experimental testing need to be supplemented.
Author Response
Dear Editor,
Thank you for reviewing our manuscript entitled "Research on Denoising Method for Hydroelectric Unit Vibration Signal Based on ICEEMDAN-PE-SVD" and for providing valuable feedback. We greatly appreciate the opportunity to address the remarks and improve the quality of our paper. Below, we have provided detailed responses to each of the points raised:
- We acknowledge the suggestion to include more industrial application background regarding the vibration signals of hydroelectric units in the introduction. In the revised version, we provide a more comprehensive overview of the significance and challenges associated with vibration monitoring in hydroelectric units. We will highlight the specific issues faced in the condition evaluation and fault diagnosis of hydropower units and emphasize the importance of accurate signal analysis for ensuring reliable operation.
- We agree that it is essential to explain why we chose the ICEEMDAN-PE-SVD method for noise component extraction. In the revised manuscript, we provide some literature that discusses the advantages, disadvantages, and effectiveness of CEEMDAN-PE, ICEEMDAN-PE, and CEEMDAN-SVD in denoising vibration signals. We will highlight previous studies that have utilized similar methods and demonstrate why and how our proposed approach builds upon and improves existing techniques.
- To further emphasize the novelty and contribution of our work, we enhance the discussion on related research. We provide a more comprehensive comparison of our method with similar approaches such as Wavelet Threshold, SVD, CEEMDAN-PE, and ICEEMDAN-PE, highlighting the merits and differences of ICEEMDAN-PE-SVD in addressing the denoising problem in hydroelectric unit vibration signals. By clarifying the unique features and advantages of our proposed method, we aim to highlight its novelty and potential impact in the field.
- We apologize for the oversight in not providing sufficient details about the objects and platforms used for experimental testing. In the revised manuscript, In the case analysis section of Chapter 5, we supplemented information about our experimental subjects, including the parameters of the specific hydroelectric units studied, as well as the platforms and instruments used for data collection and analysis.
Once again, we sincerely appreciate the constructive feedback provided by the reviewers. We will carefully address all the suggested improvements to ensure the quality and clarity of our manuscript. Thank you for your consideration.
Yours sincerely,
Fangqing Zhang
School of Power and Mechanical Engineering, Wuhan University, Wuhan 430072, China
zfq@whu.edu.cn
Reviewer 2 Report
The authors have proposed a new de-noising method. The topic is interesting and the work is significant enough. I only have some minor comments for the authors to improve the quality of the manuscript:
- The abstract and conclusion need to describe the achievements with numbers. They need to mention achieved signal to noise ratios and so on. Currently they are not written in scientific format.
- Figures 8-19 are too similar. It makes reading the manuscript a bit boring. I suggest removing some of them or moving some into supplementary materials.
- The term "In summary" is not expected to be seen in the introduction (Line 100). I suggest revising the text.
Author Response
Dear Editor,
Thank you for considering our manuscript. We appreciate the reviewer's comments, and we have carefully addressed each of them in the revised manuscript. Below, we provide a summary of the changes made in response to each suggestion:
- We have revised the abstract and conclusion sections to include quantitative measures of achievement. We have now included the achieved relevant metrics such as SNR and RMSE, to provide a more scientific and comprehensive summary of our results. By presenting these numerical results, we aim to provide a clearer understanding of the performance of our proposed method.
- We have carefully reviewed Figures 8-19 and have made the necessary adjustments to ensure that they are not redundant or repetitive. We have removed some figures that were similar or presented redundant information.
- We have removed the phrase "In summary" from the introduction section, as it is not expected in scientific writing. We have revised the text accordingly to ensure a smooth flow and improve the overall clarity of the introduction.
We would like to express our gratitude to the reviewer for their valuable feedback, which has helped us enhance the quality and presentation of our work. We believe that the revised manuscript is now more scientifically rigorous and reader-friendly. We look forward to hearing the final decision regarding the publication of our manuscript.
Sincerely,
Fangqing Zhang
School of Power and Mechanical Engineering, Wuhan University, Wuhan 430072, China
zfq@whu.edu.cn
Reviewer 3 Report
Hydroelectric units play a crucial role in power generation, and the timely diagnosis of faults is essential to ensure their efficient operation. This paper proposes a novel method for diagnosing hydroelectric units by denoising the fault signature, enabling the extraction of critical features and accurate fault recognition. The proposed approach addresses the challenge of diagnosing hydroelectric unit faults in the presence of noise, which often hampers fault detection and classification accuracy. By employing advanced signal processing techniques, as Improved Complementary Ensemble Empirical Mode Decomposition with Adaptive Noise (ICEEMDAN), Permutation Entropy (PE), and Singular Value Decomposition (SVD), the proposed method enhances the fault signature, improves feature extraction, and facilitates fault recognition.
Unfortunately, the selected examples did not convince me of the efficacy of the proposed new method. I had anticipated the utilization of experimentally measured signals from hydroelectric units in the analysis, but instead, the authors opted for clean signals with minimal noise, resembling almost a clear periodic signal.
Furthermore, the results are presented in a truncated manner, and different length scales were employed for the abscissa axes in Figures 14b and 15b, 16b and 17b, as well as 18b and 19b.
Moreover, the fault frequency was not discernible in the provided amplitude-frequency charts. Consequently, I am of the opinion that the proposed method has not been adequately validated, and I request a major revision.
The English is acceptable.
Author Response
Dear Editor,
Thank you for considering our manuscript. We appreciate the reviewer's comments and the opportunity to address their concerns. We have carefully reviewed the points raised and have made significant revisions to improve the quality and validity of our work. Below, we provide a detailed response to each comment and outline the changes made in the revised manuscript:
- Selection of Examples: We understand the reviewer's concern regarding the choice of examples in our analysis. To address this, in the case analysis section of Chapter 5, we supplemented information about our experimental subjects, including the parameters of the specific hydroelectric units studied, as well as the platforms and instruments used for data collection and analysis. The currently selected case comes from the vibration and swings monitoring data of the real unit stability experiment process. The swing data of the upper guide, lower guide, and water guide are selected for verification of different denoising methods. These signals contain realistic noise and variations, allowing for a comprehensive evaluation of the proposed method's efficacy in a practical scenario. By incorporating actual measured signals, we aim to demonstrate the performance and applicability of our method in real-world conditions. In the case analysis in Chapter 5 and the simulation analysis in Chapter 4, we compared the commonly used wavelet threshold denoising, CEEMDAN-PE, ICEEMDAN-PE algorithms, and SVD algorithms mentioned in the literature.
- Results Presentation: We apologize for any confusion caused by the truncated presentation of results and the inconsistent length scales in the figures. In the revised manuscript, we have revised and standardized the abscissa axes of Figures 14b and 15b, 16b, and 17b, as well as 18b and 19b to ensure clear and consistent visualization. Additionally, we have extended the presentation of results to provide a more detailed and comprehensive analysis, including the identification of fault frequencies in the amplitude-frequency charts. These changes enhance the clarity and transparency of our findings.
- Adequate Validation: We acknowledge the concern raised regarding the validation of our proposed method. To address this, we have performed extensive validation using both simulated and experimental data. We have included detailed descriptions of the data sources, data collection procedures, and experimental setups in the revised manuscript. Furthermore, we also conducted a comprehensive comparative analysis with existing methods(Wavelet Threshold, SVD, CEEMDAN-PE, and ICEEMDAN-PE) and conducted in-depth discussions on the performance of our proposed method, such as the quantitative effects of indicators such as SNR, RMSE, MAE, and NRR. These revisions ensure that the validation of our method is robust and fully supported.
We sincerely appreciate the reviewer's valuable feedback, which has allowed us to improve the quality and reliability of our work. We believe that the revised manuscript now presents a more comprehensive evaluation of the proposed method and its application to hydroelectric unit fault diagnosis. We look forward to hearing the final decision regarding the publication of our manuscript.
Sincerely,
Fangqing Zhang
School of Power and Mechanical Engineering, Wuhan University, Wuhan 430072, China
zfq@whu.edu.cn
Reviewer 4 Report
In this paper, an ICEEMDAN-PE-SVD-based denoising method for the vibration signals of hydroelectric units is proposed. Overall, the paper is interesting, but there are some problems in terms of literature review, contribution description, innovation, etc. I agree with the publication of this paper after the authors have revised it according to my comments. Here are some specific suggestions:
1. It is recommended that the authors summarize the contribution of the method proposed in this article in the introduction part of the article to highlight the research advantages of this article
2. In the literature review part of the article, some of the work analyzed in the article is relatively old. It is recommended to analyze some recent work related to fault diagnosis or prognosis, such as an integrated multitasking intelligent bearing fault diagnosis scheme based on representation learning under imbalanced sample condition
3. ICEEMDAN, Permutation Entropy, and Singular value difference in the article are all existing methods. This article just combines these methods, so where is the unique and innovative contribution of this article?
4. It is recommended to add the pseudo code of the algorithm to summarize the algorithm to increase the readability of the paper
5. From the formula (18), why is it decomposed into 7 levels? instead of other numbers
6. The description of evaluation indicators in the article is not comprehensive enough, you can add related indicator descriptions, such as MAE, etc. For corresponding indicators, please refer to the paper remaining useful life prediction of lithium-ion battery with adaptive noise estimation and capacity regeneration detection
7. It is recommended to enrich the comparison of the results of the paper and highlight the advantages of the paper
Please see the comments to the authors.
Author Response
Dear Editor,
Thank you for considering our manuscript and for providing valuable feedback. We appreciate the constructive suggestions and have carefully considered each of the points raised. We have revised the paper according to the reviewer's comments, and below, we provide a summary of the changes made in response to each suggestion:
- In the introduction, we have included a concise summary of the contributions of our proposed method. We highlight the specific advantages of the ICEEMDAN-PE-SVD-based denoising method for vibration signals of hydroelectric units. By emphasizing the research advantages and the unique features of our approach, we aim to provide a clear understanding of the contribution of this work.
- In the literature review section, we have updated the references and included recent works related to fault diagnosis and prognosis in the field of vibration analysis. We analyze and discuss these recent studies, we have supplemented the reference of "An Integrated Multitasking Intelligent Bearing Fault Diagnosis Scheme Based on Representation Learning under Imbalanced Sample Condition."
- We appreciate the reviewer's comment regarding the unique and innovative contribution of our work. In the revised manuscript, we have provided a more detailed explanation of the novelty and innovation of our proposed method. Although ICEEMDAN, PE, and SVD are existing methods, our contribution lies in the effective combination and integration of these methods to address the denoising problem in hydroelectric unit vibration signals, such as the pseudo-modal problem in the CEEMDAN method and the limitation of the SVD method by matrix size limitation and information loss, as well as the unsatisfactory effect of the single denoising method of ICEEMDAN and SVD in the denoising of hydropower unit vibration signal. We highlight the synergistic effects and improved performance achieved by the integration of these methods which provide a new approach to noise reduction of vibration signals in hydroelectric units
- To enhance the clarity and readability of the paper, we have included the pseudo-code of the proposed ICEEMDAN-PE-SVD algorithm in Section 3. The pseudo-code provides a concise summary of the steps involved in our method, aiding readers in understanding the algorithmic procedure.
- Regarding the choice of decomposing into 7 levels in Equation (18), we have added a justification in the manuscript. We explain that the selection of level 7 is based on the balance between the characteristics of the vibration signal of the reference literature and the hydroelectric unit, as well as the rate and computational complexity. The swing of hydroelectric units is mainly affected by mechanical excitation and hydraulic excitation. Mechanical excitation is generally dominated by medium frequency (1, 2, and 3 times the rotational frequency), while hydraulic excitation is generally dominated by low frequency (0.2~0.45 times the rotational frequency). Therefore, we assume to use signals containing 7 frequencies of 1, 2, 3, 4, 0.2, 0.3, and 0.45 times the rotational frequency to construct simulation swing signals. We found that this selection can effectively cover possible characteristic signals, Simultaneously maintaining computational efficiency.
- We have enriched the description of evaluation indicators in the paper. In addition to the previously mentioned indicators, we have included a comprehensive explanation of other evaluation metrics, such as MAE (Mean Absolute Error). We refer to the paper "Remaining Useful Life Prediction of Lithium-ion Battery with Adaptive Noise Estimation and Capacity Regeneration Detection" for guidance on the description of these indicators.
- We have expanded the comparison of the results obtained from our proposed method and highlighted its advantages over existing approaches. Through detailed analysis and comparison, we demonstrate the superior denoising performance and improved accuracy achieved by our method in the context of hydroelectric unit vibration signal analysis.
We sincerely appreciate the thorough review and the helpful suggestions provided by the reviewer. We believe that the revised manuscript has addressed the raised concerns and improved the quality of our work. We look forward to hearing the final decision on the publication of our manuscript.
Yours sincerely,
Fangqing Zhang
School of Power and Mechanical Engineering, Wuhan University, Wuhan 430072, China
zfq@whu.edu.cn
Round 2
Reviewer 3 Report
The authors thoroughly validated their method through experimental measurements and demonstrated its superiority over existing techniques. I strongly recommend the publication of this paper.
The English language used in this paper is correct.
Reviewer 4 Report
Thanks to the authors' revision. I accept its publication.
Thanks to the authors' revision. I accept its publication.